# Characterization of the IAA-Producing and -Degrading *Pseudomonas* Strains Regulating Growth of the Common Duckweed (*Lemna minor* L.)

**DOI:** 10.3390/ijms242417207

**Published:** 2023-12-06

**Authors:** Tatjana Popržen, Ivan Nikolić, Dijana Krstić-Milošević, Branka Uzelac, Milana Trifunović-Momčilov, Marija Marković, Olga Radulović

**Affiliations:** 1Department of Plant Physiology, Institute for Biological Research “Siniša Stanković”—National Institute of the Republic of Serbia, University of Belgrade, 142 Bulevar Despota Stefana Street, 11060 Belgrade, Serbia; tatjanap@live.com (T.P.); dijana@ibiss.bg.ac.rs (D.K.-M.); branka@ibiss.bg.ac.rs (B.U.); milanag@ibiss.bg.ac.rs (M.T.-M.); marija.nikolic@ibiss.bg.ac.rs (M.M.); 2Center for Biological Control and Plant Growth Promotion, Faculty of Biology, University of Belgrade, 16 Studentski Trg Street, 11000 Belgrade, Serbia; ivan.nikolic@bio.bg.ac.rs

**Keywords:** indole-3-acetic acid, duckweed, pseudomonas strains, plant–microbe interactions

## Abstract

The rhizosphere represents a center of complex and dynamic interactions between plants and microbes, resulting in various positive effects on plant growth and development. However, less is known about the effects of indole-3-acetic acid (IAA) on aquatic plants. In this study, we report the characterization of four *Pseudomonas* strains isolated from the rhizosphere of the common duckweed (*Lemna minor*) with IAA-degradation and -utilization ability. Our results confirm previous reports on the negative effect of IAA on aquatic plants, contrary to the effect on terrestrial plants. *P. putida* A3-104/5 demonstrated particularly beneficial traits, as it exhibited not only IAA-degrading and -producing activity but also a positive effect on the doubling time of duckweeds in the presence of IAA, positive chemotaxis in the presence of IAA, increased tolerance to oxidative stress in the presence of IAA and increased biofilm formation related to IAA. Similarly, *P. gessardii* C31-106/3 significantly shortened the doubling time of duckweeds in the presence of IAA, while having a neutral effect in the absence of IAA. These traits are important in the context of plant–bacteria interactions and highlight the role of IAA as a common metabolite in these interactions, especially in aquatic environments where plants are facing unique challenges compared to their terrestrial counterparts. We conclude that IAA-degrading and -producing strains presented in this study might regulate IAA effects on aquatic plants and confer evolutionary benefits under adverse conditions (e.g., under oxidative stress, excess of IAA or nutrient scarcity).

## 1. Introduction

Indole-3-acetic acid (IAA), the major phytohormone active in a variety of physiological processes in plants, is produced not only by plants but also by various microorganisms, including bacteria [1,2,3]. In fact, some bacteria are extremely efficient in IAA production to the extent that they can exert a massive negative or positive influence on plant growth, e.g., in pathogenesis or in enhancing the biomass production and resistance of the plants [4,5]. While the effects of bacterial IAA on the physiology of terrestrial plants are well described, little is known about its effects on aquatic plants [3]. Only in recent decades, the interactions between bacteria and aquatic plants and the importance of these interactions in water ecosystems were studied in greater detail [6]. The crucial role of IAA is underlined by the fact that more than 80% of the bacteria from the rhizosphere produce IAA [1,7]. The rhizosphere is the area in the proximity of plant roots that is defined by the metabolic activity of the plants and is inhabited by various microorganisms that interact with the plants [8]. Furthermore, the root produces a large portion of total IAA in plants and is also particularly sensitive to IAA, responding by morphogenetic changes like lateral root growth or the termination of root elongation [9,10]. However, due to their regressive evolution, the roots of duckweeds do not produce lateral roots at all, and it is even reported that the entire organ is practically obsolete, as at least some species of Lemnaceae can thrive even when their roots are mechanically removed [11]. Therefore, due to their morphological and evolutionary uniqueness, it is necessary to examine the morphological and physiological changes in response to IAA in duckweeds. Furthermore, since aquatic plants are not affected only by the IAA of their own making, it is important to examine the effects of IAA of microbial origin. In this study, our focus will be the IAA produced by bacteria.

Bacterial IAA is not just a product of bacterial metabolism, as previously assumed, but an active compound that acts as a signaling molecule and is recognized by both bacteria and plants [12,13,14]. In fact, IAA and similar indole compounds appear to be a universal signal in the animal kingdom that is involved in interspecies communication, even between taxa that are evolutionarily very distant [14]. It is becoming increasingly apparent that IAA positively influences the stress response of not only plants but bacteria as well [15]. Furthermore, IAA acts as an inter-bacterial signal for the initiation of biofilm formation [16,17]. Biofilm is a complex form of bacterial organization that ensures higher colony survival, allows for the exchange and concentration of nutrients and signaling molecules and provides physical support and protection for individual bacteria [18]. Rhizospheric bacteria mostly exist as sessile organisms in biofilms on the surface of plants, and thus they provide additional protection to plant roots against changing and unfavorable external conditions [18].

The common duckweed (*Lemna minor*) is a small aquatic plant of simplified morphology, characterized by cosmopolitan distribution, rapid vegetative reproduction and adaptability to very hostile environments. Duckweeds coexist with a wide range of different microorganisms, most notably bacteria. In fact, duckweeds possess a remarkable tolerance to the high density of various microorganisms, due in large part to their unique innate defense system: this makes them a particularly flexible model organism for the study of bacteria–plant interactions [19]. Some of these bacteria are identified as plant-growth promoting (PGPB); in our previous work, we reported seven bacterial strains that were, in addition to removing phenol, also able to synthetize IAA [20]. We also reported the domination of *Pseudomonas* strains among the rhizosphere-associated isolates: almost half of the identified bacterial strains belonged to the *Pseudomonas* genus [21]. Bacterial genus *Pseudomonas* is one of the largest Gram-negative genera [22]. This genus consists of highly versatile and metabolically active microorganisms, which can remove a wide range of toxic compounds using various biological mechanisms [23] and are reported to possess PGP properties, including IAA production [24]. Representatives of this genus can be found in virtually every ecosystem. In the rhizosphere, they play an important role in removing noxious organic compounds and metals, thus ensuring that the plants are protected from their toxic effects [22]. Moreover, pseudomonads actively synthetize various metabolites that can be utilized by both bacteria and plants [25]. Given the cosmopolitan distribution of duckweeds, their tolerance to different bacteria and the ability of bacteria associated with the duckweeds’ rhizosphere, particularly those from the Pseudomonas genus, to remove toxic compounds, these small aquatic plants, when inoculated with selected strains with desirable properties, can be widely used in various biotechnological contexts and are therefore used as a dual model system in this study [23,24].

In this study, we analyzed the ability of bacteria isolated from the rhizosphere of an aquatic plant (*L. minor*) to degrade and produce IAA and to utilize IAA as a sole carbon source. Since IAA is an ancient and ubiquitous signaling molecule, we also analyzed phylogenetic relations between bacterial strains that were identified as producers and degraders of IAA. To ascertain if IAA is beneficial to selected bacteria, we also tested the biofilm formation and viability of these bacterial strains under oxidative stress and whether IAA had a positive effect on their survival. Finally, we monitored the effect of selected bacterial strains on the multiplication of duckweeds in vitro. Our aim in this study was to identify IAA-synthetizing and -degrading bacterial strains that are naturally associated with the duckweeds, using duckweeds as a model organism, and to elucidate some aspects of the interplay between plants, bacteria and their common metabolite, IAA.

## 2. Results

### 2.1. Analysis of Bacterial Ability to Utilize IAA as A Sole Carbon Source and Evolutionary Relations between IAA-Producing and -Utilizing Bacterial Strains

To ascertain which strains from a previously reported collection [21] produced IAA, the microwell screening method was used. All tested strains produced detectable amounts of IAA in the range from 2.4 to 20.8 mg L^−1^. The least amount was produced by *Enterobacter* sp. D5-1-102 (ID 6), and the greatest amount was produced by *Pseudomonas yamanorum* C44-104/1 (ID 47) (Figure 1). Strains that produced amounts of IAA equal to or greater than the reference strain, *P. oryzihabitans* D1-104/3, were tested for their ability to utilize IAA as the sole carbon source (Figure 1). The reference amount was (10.2 ± 0.3) mg L^−1^. In total, 11 strains produced IAA equal to *P. oryzihabitans* D1-104/3 (ID 15) or more.

Changes in the turbidity of the minimal M9 medium with IAA (20 mg L^−1^) as the sole carbon source were used as evidence of successful utilization of IAA. Based on these initial experiments (IAA production and ability to use IAA as the sole carbon source), four strains were selected. These four strains were *Pseudomonas oryzihabitans* D1-104/3, *Pseudomonas putida* A3-104/5, *Pseudomonas gessardii* C31-106/3 and *Pseudomonas yamanorum* C44-104/1 (Table 1).

Growth curves were constructed for these IAA-producing, IAA-utilizing strains growing in the minimal M9 medium with IAA as the sole carbon source (Figure 2A). The fastest growth was observed in *P. putida* A3-104/5 while no growth was detected in *P. gessardii* C31-106/3 (Figure 2A). Simultaneously, samples for the detection of IAA were taken, and degradation curves were constructed for each strain. After 4 h of incubation, *P. oryzihabitans* D1-104/3 decreased the amount of initial IAA by 20% while other bacterial strains had an increase of approximately 10%. After 8 h of incubation, there was a 50% decrease in initial IAA in all samples (except for *P. gessardii* C31-106/3). After 12 h of incubation, there was no difference in the amount of degraded IAA between the strains, and 60% of the initial IAA was degraded by every strain (Figure 2B). As expected, the amount of IAA in the sample inoculated with *P. gessardii* C31-106/3 was comparable to the sterile control.

Generation times, growth rates and relative yields were calculated for all strains grown in the M9 minimal medium with IAA, except for *P. gessardii* C31-106/3 (Table 2). Unlike the minimal M9 medium with IAA as the sole carbon source, *P. gessardii* C31-106/3 was successfully grown in the LB medium, although with a longer generation time compared to other strains (Appendix A).

Partial 16S rDNA sequences were used to construct phylogenetic trees for four selected bacterial strains (*P. oryzihabitans* D1-104/3, *P. putida* A3-104/5, *P. gessardii* C31-106/3 and *P. yamanorum* C44-104/1) and twelve evolutionary closely related taxa whose sequences were taken from Genbank. The optimal tree is shown in Figure 3. There were a total of 1538 positions in the final dataset.

### 2.2. Bacterial Viability after H_2_O_2_ and IAA Treatment

To obtain information about the potentially protective effect of IAA on oxidative stress in bacterial strains, bacterial strains were cultivated in an IAA-supplemented liquid LB medium and then treated with H_2_O_2_, as previously described. The bacterial growth was observed on the Petri dish, and the growth of cultures exposed to oxidative stress after cultivation in an IAA-supplemented medium was compared with cultures grown in a medium without IAA. The addition of IAA induced the survival of *P. putida* A3-104/5 after exposure to H_2_O_2_ (Figure 4). In *P. gessardii* C31-106/3, although exogenous IAA led to a significant increase in the number of untreated colonies, it had no effect on the survival of oxidative stress. After the treatment with H_2_O_2_, there were no detectable *P. yamanorum* C44-104/1 colonies, regardless of previous cultivation with IAA.

### 2.3. Biofilm Formation and Chemotaxis

To evaluate the colonization capabilities of tested *Pseudomonas* strains, we analyzed biofilm formation in the microtiter plate. The biofilm formation of *P. putida* A3-104/5 was stimulated in the presence of 10 µg L^−1^ of IAA, while it was apparently inhibited in *P. oryzihabitans* D1-104/3 and in *P. gessardii* C31-106/3, with no effect on *P. yamanorum* C44-104/1 (Figure 5A). The addition of 10 mg L^−1^ of IAA led to a statistically significant increase in biofilm formation only in *P. yamanorum* C44-104/1, whereas in the remaining strains, it led to no significant difference (*P. putida* A3-104/5) or a reduction in biofilm formation (*P. oryzihabitans* D1-104/3 and *P. gessardii* C31-106/3).

After 24 h of incubation at +30 °C, all bacterial strains, except for *P. gessardii* C31-106/3, showed radial expansion in an IAA-supplemented M9 medium. Radial expansion was significantly increased in *P. putida* A3-104/5 in the IAA-supplemented medium. In agar plates supplemented with IAA, *P. putida* A3-104/5 formed a halo whose diameter was (11 ± 1) mm on average. In fact, the radial expansion of all strains was positively affected by IAA, except for *P. gessardii* C31-106/3 where radial expansion was comparable to soft agar without supplementation (Figure 5B). The applied concentration of sodium benzoate apparently had an inhibitory effect on all tested strains, and it significantly limited the expansion of strains *P. oryzihabitans* D1-104/3, *P. putida* A3-104/5 and *P. yamanorum* C44-104/1 (Figure 5B). Expansion halos in agar plates supplemented with sodium benzoate were reduced compared to negative control plates (plates without supplementation), with an average diameter of (3.25 ± 0.48) mm in sodium benzoate agar and (5.75 ± 0.95) mm in non-supplemented agar, suggesting that sodium benzoate had an inhibitory effect on the strains at the applied concentration.

### 2.4. HPLC-DAD Determination of IAA

To confirm the presence of IAA in the supernatant of selected strains, high-performance liquid chromatography (HPLC) was performed. The peak of identified IAA from the supernatant of referent strain *P. oryzihabitans* D1-104/3 appears after around 21 min, as a peak of about 90 mAU. A representative chromatogram of ethyl-acetate extract derived from the supernatant of *P. oryzihabitans* D1-104/3 culture is presented in Figure 6.

The calculated content of IAA, expressed in milligrams per gram of dry extract or milligrams per OD_600_ unit, is presented in Table 3.

### 2.5. Co-Cultivation of Bacteria and Duckweeds

Without IAA, statistically significantly longer DT had duckweeds inoculated with *P. oryzihabitans* D1-104/3 and *P. yamanorum* C44-104/1 compared to control without IAA (Appendix A). With IAA, statistically significantly shorter DT had duckweeds inoculated with *P. putida* A3-104/5, *P. gessardii* C31-106/3 and *P. yamanorum* C44-104/1 compared to the control with IAA. Furthermore, in the presence of IAA, the DT of control duckweeds was almost doubled compared to control duckweeds without IAA. In fact, duckweeds inoculated with three tested bacterial strains had significantly longer DT in the presence of IAA compared to inoculated duckweeds without IAA, except for duckweeds inoculated with *P. yamanorum* C44-104/1 (Figure 7C).

## 3. Discussion

### 3.1. Bacterial Production of IAA

IAA is considered one of the most important phytohormones. It has been proven that IAA-producing bacteria have beneficial effects on plant growth, which is why the analysis of the production of this hormone is an important indicator [26]. However, the effect of exogenous plant growth regulators on plant growth and biomass production was rarely investigated, especially in aquatic plants [27]. Therefore, our study was focused on investigating these effects. After analyzing our previously reported collection of 37 bacterial strains associated with the rhizosphere of *L. minor* [21], 12 bacterial strains (including the reference strain) were found to produce equal or higher quantities of IAA than the reference strain *P. oryzihabitans* D1-104/3, even without exogenous tryptophan (L-Trp) in the nutrient medium as a precursor for the biosynthesis of IAA. Although the LB medium contains L-Trp, this amino acid is prone to degradation, especially to photodegradation—the estimated amount of L-Trp is small (less than 1 mM) and variable in LB media [28]. Therefore, even in the presence of a very small amount of L-Trp, bacterial strains from our collection were able to produce detectable quantities of IAA. Further analyses showed that, out of these IAA-producing strains, four can also utilize IAA as the sole carbon source. Interestingly, all four strains with these characteristics belong to the *Pseudomonas* genus, and it was precisely in another *Pseudomonas* strain, namely *Pseudomonas putida* 1290, that the gene cluster for IAA metabolism was discovered [29]. Additionally, when exogenous tryptophan was present, the amount of IAA produced by two out of four of these strains (*P. oryzihabitans* D1-104/3 and *P. putida* A3-104/5) was increased at least three times, indicating that their IAA production is stimulated (upregulated) in the presence of tryptophan (Appendix A). The ability to produce IAA even when tryptophan is very limited and the ability to utilize IAA as the only energy source make these four strains particularly interesting for future research and potential biotechnological applications, e.g., in agriculture [24]. The species *P. oryzihabitans* was taken as a reference for IAA production in this study since it is a PGP bacterium with a previously reported and well-described IAA-producing ability [30,31]. The HPLC analysis of IAA production in the LB medium with exogenous L-Trp confirmed that this strain was exceptionally efficient in producing IAA compared to other analyzed strains. A phylogenetic tree was organized into two clusters, reflecting genetic relationships between the strains: one cluster contained *P. oryzihabitans* D1-104/3 and *P. gessardii* C31-106/3, while the other contained *P. putida* A3-104/5 and *P. yamanorum* C44-104/1. Phylogenetically, *P. oryzihabitans* D1-104/3 and *P. putida* A3-104/5 are closer to *P. putida* 1290 than *P. yamanorum* C44-104/1 and *P. gessardii* C31-106/3. In fact, *P. gessardii* C31-106/3 is genetically the most diverse compared to other *Pseudomonas* strains in this study, at least according to the results of phylogenetic analysis conducted in this research. The genetic divergence might be correlated with some observed physiological differences of this strain (e.g., inability to grow at 5 mM IAA, lack of radial expansion, overall slow growth). Hypothetically, the divergent evolution of this strain, reflected in 16S rRNA gene sequences, might have led to the development of physiological differences observed in this study. However, further studies of the phylogeny of these strains are necessary to form any solid conclusion.

### 3.2. Bacterial Degradation of IAA

Although four bacterial strains tested in this study were grown successfully on minimal M9 agar with 20 mg L^−1^ IAA, one strain (*P. gessardii* C31-106/3) grew poorly in the liquid M9 medium with 5 mM of IAA (~1 g L^−1^) as the sole carbon source. It is also worth noting that this bacterium had a longer generation time even in the complex nutrient LB medium, suggesting that its requirements for optimal growth may be specific for this strain compared to others. It is also possible that this concentration of IAA was toxic to this strain or that this strain, like some other bacteria, efficiently degrades IAA only under certain conditions [9]. Three remaining strains could utilize IAA under test conditions. Compared to *P. putida* 1290, these three strains had at least three times longer generation times [9]. This underlines the genetic and physiological differences between these phylogenetically close bacterial strains. It is worth noting that, although it has been known for decades that IAA-producing bacteria are very common in the rhizosphere [7], bacteria with the ability to both degrade and produce IAA are seemingly very rare and only occasionally reported in the scientific literature [16,32,33,34]. Even less is known about their presence and role in the rhizosphere of aquatic plants. In terrestrial plants, the role of the dual ability to produce and degrade IAA is associated with the colonization of plant cells as endophytes and to form nodules (examples being the nodule forming *Bradyrhizobium japonicum* in soybean or the endophyte *Burkholderia pyrrocinia* in poplar) or infect the plant as pathogens (*P. savastanoi* in various plant species) [5,33,35,36]. The degradation and production of IAA probably confer an evolutionary advantage in competition with other microorganisms inhabiting the rhizosphere, since IAA is an important, ubiquitous molecule controlling various physiological processes in various, even evolutionary distant, organisms [10]. Furthermore, it was reported that IAA has different physiological effects on aquatic plants compared to terrestrial ones [3]. Unlike bacterial IAA biosynthesis, IAA biodegradation is still insufficiently investigated [10]. It is possible that the dual ability to control IAA by degradation and biosynthesis helps bacteria to successfully colonize the rhizosphere, which is especially important in aquatic ecosystems where nutrient availability is very limited and prone to loss through dispersion and dilution.

### 3.3. IAA and Bacterial Viability under Oxidative Stress, Biofilm Formation and Chemotaxis

Exogenous IAA influences a wide range of physiological processes in bacteria, including tolerance to stress and biofilm formation [15,16,17,18]. In our study, the effects of IAA on bacterial viability, motility and biofilm depended on the concentration and type of bacterial strain. *P. oryzihabitans* D1-104/3 and *P. gessardii* C31-106/3 survived oxidative stress regardless of whether IAA was present in the nutrient medium or not, although there was a decrease in the number of viable colonies in both cases. The most striking result is that *P. putida* A3-104/5 cultures that were grown in an IAA-supplemented medium formed viable colonies after oxidative stress treatment. However, no viable colonies of *P. putida* A3-104/5 were detected if no prior cultivation in the IAA-supplemented medium occurred. It seems that, for this strain, IAA does have a positive effect in terms of resistance to oxidative stress. In contrast, in the case of *P. yamanorum* C44-104/1, no colonies were detected even when cultures were grown in the IAA-supplemented medium. Regarding biofilm formation, biofilm was quantifiable in all four strains. However, biofilm formation was dose-dependent and differed between the strains: IAA stimulated the biofilm formation of *P. putida* A3-104/5 (at 10 µg L^−1^) and *P. yamanorum* C44-104/1 (at 10 mg L^−1^). Simultaneously, IAA at both concentrations had a negative effect on biofilm formation in *P. oryzihabitans* D1-104/3 and *P. gessardii* C31-106/3. Biofilms are particularly important in aquatic environments as they concentrate nutrients that would otherwise be dispersed in water and offer physical support and protection to bacteria and plant roots [17,18]. Furthermore, IAA had a differing effect on bacterial motility, as well. Radial expansion was stimulated in the presence of IAA: this effect was significant in three out of four strains, with *P. gessardii* C31-106/3 being the only exception. It is worth noting that not all bacteria can simultaneously degrade and recognize the same compound as a chemoattractant [37]. This interesting characteristic of the three strains reported in this study might be explained by the bacterial ability to recognize IAA of plant origin as a signal for colonization and subsequent transformation into chemical energy, which is particularly important for colonization in aquatic, nutrient-poor environments. *P. gessardii* C31-106/3 did not show radial expansion under these experimental conditions, which is in line with our previous observations regarding IAA toxicity and this strain. Unlike many other *Pseudomonas* strains, e.g., *P. putida* 1290 and *P. putida* PRS2000, benzoate was not a chemoattractant for strains in this study, hinting again at possible genetic differences between these phylogenetically close taxa [37,38]. Transcriptional analysis might reveal different patterns of gene expression in these strains that could explain differential responses, especially in terms of overall viability under oxidative stress and biofilm formation. Additional testing of chemotactic behavior, e.g., with different well-known chemoattractants and with a microfluidic assay, would also help understand the IAA-driven motility of these strains.

### 3.4. Effects of Bacterial Strains and Exogenous IAA on Doubling Time of Duckweeds

The effects of bacteria on the growth of duckweeds depended on exogenous IAA and the strain of bacteria. Exogenous IAA exerted significant negative effects on duckweeds. This is in accordance with findings from previous studies which report that IAA has either a negative or a neutral effect on the growth of duckweeds [3,27]. All strains, except for *P. oryzihabitans* D1-104/3, shortened the doubling time of duckweeds grown in an IAA-containing medium, suggesting that they somewhat ameliorated the toxic effects of IAA, probably by removing excess IAA from the medium. For comparison, Leveau and Lindow (2005) reported that, while *P. putida* 1290 on its own had no effect on the root development of horse radish, it completely abolished the negative effects of 1 mM (≈175 mg L^−1^) of IAA while 1 µM of IAA stimulated the positive effects of this bacterium [9]. This again emphasizes the differences in the effects of IAA-degrading/producing bacteria on aquatic versus terrestrial plants. Moreover, it should be noted that *P. putida* A3-104/5 and *P. gessardii* C31-106/3 significantly alleviated the effects of IAA while also having a neutral effect on the doubling time of duckweeds in the absence of IAA. Interestingly, *P. oryzihabitans* D1-104/3 increased the doubling time of duckweeds grown in the medium without IAA and had no effect on duckweeds grown in an IAA-supplemented medium. It is worth noting that *P. oryzihabitans* is an endophyte; however, whether this particular strain used in this study is one is yet to be determined. Furthermore, under experimental conditions in this study, *P. oryzihabitans* D1-104/3 showed no overt phytopathogenic traits (in terms of pathological changes like necrosis or wilting). The fact that this strain increased doubling time (slowed the multiplication of duckweeds) can be attributed to multiple factors, including its production of IAA which, as stated, has a negative effect on the multiplication of duckweeds and apparently overrode the positive effects of IAA degradation in this case. Even though the positive effects of bacteria can be also attributed to the surplus of nutrients released by dead bacterial cells, the differences in duckweeds’ doubling times dependent on bacterial strain suggest that their effects are more complex. However, without bacterial mutants with impaired or stimulated production of IAA, or with impaired degradation of IAA, it is impossible to connect the effects of these strains on duckweeds with the regulation of IAA levels. Genomic and metabolomic analyses are therefore necessary to gain a clear perspective on the physiological and genetic diversity of these strains and how their evolution led to biosynthesis and biodegradation pathways for IAA. However, the results presented in this initial study seem promising and will prompt future research in new directions.

## 4. Material and Methods

### 4.1. Plant Material and Growth Conditions

Duckweed (*L. minor*) was obtained from the nursery garden of the Institute for Biological Research “Siniša Stanković”, University of Belgrade, and kept in clear glass containers filled with room-temperature, non-sterile tap water. Prior to conducting experiments, plants were sterilized using H_2_O_2_ as a sterilizing agent and kept in Murashige and Skoog medium supplemented with 30 g L^−1^ sucrose. Plants were grown at 24 ± 2 °C (under fluorescent light of 40 μmoL m^−2^ s^−1^ with 16 h light/8 h dark photoperiod).

### 4.2. Initial IAA Screening of Bacterial Isolates Associated with the Rhizosphere of L. minor: The Microwell Method

Bacterial strains previously isolated from the rhizosphere of *L. minor* [21] were tested for IAA production by incubating in Luria-Bertani (LB) medium, at +28 °C. No exogenous tryptophan was added as a precursor for IAA. After 24 h of incubation, suspensions were briefly centrifuged, and supernatants were mixed with Salkowski reagent (prepared by mixing 1 mL of 0.5 M FeCl_3_ solution with 100 mL of 35% perchloric acid) in 1:1 ratio, in a microwell plate. The plate was incubated at room temperature, in dark, for 30 min. Absorbance of colored tris-(indole-3-acetato)iron(III) complex was measured at 536 nm, on Agilent 8453 UV/Visible Spectrophotometer.

### 4.3. Bacterial Growth in Minimal M9 Medium with IAA as the Sole Carbon Source

Isolates that were identified with the microwell method as IAA producers with ability to form IAA equal to or greater than that of the reference strain (*P. oryzihabitans* D1-104/3) were tested for their ability to grow on minimal M9 agar with 20 mg L^−1^ of IAA as the sole carbon source. Bacteria were incubated at +30 °C over a period of 3 days. Only bacterial strains with the ability to form colonies on M9 agar with IAA as the sole carbon source were selected for further experiments. Overnight cultures of selected strains were centrifuged briefly, to remove LB medium. Pellets were washed in M9 medium and then transferred into flat bottom glass flasks with 50 mL of M9 minimal medium with IAA as the sole carbon source. Samples were taken at 0, 2, 4, 6, 8, 10 and 12. To compare growth parameters in a minimal nutrient medium to complex nutrient medium, bacteria were also grown in LB medium (see: Appendix A). All measurements were performed at 600 nm wavelength, on Agilent 8453 UV/Visible Spectrophotometer (Agilent Technologies, Santa Clara, CA, USA). Results of spectrophotometric measurements were presented graphically. Generation time, relative yield and growth rates were calculated in Microsoft Office Excel (2010).

### 4.4. Bacterial Degradation of IAA

Degradation of IAA was measured for 12 h, every two hours (0, 2, 4, 6, 8, 10 and 12 h) using Salkowski method with some modifications [9]. Briefly, from 1 mL of samples, bacterial pellets were removed by centrifugation, and 50 µL of supernatant was diluted with 450 µL of phosphate buffer. This dilution (60 µL) was then further diluted with 440 µL of phosphate buffer, mixed with 500 µL of Salkowski reagent (0.5 M FeCl_3_ in 35% perchloric acid) and incubated in the dark, at room temperature, for 30 min. Sterile M9 medium with 5 mM IAA was serially diluted for the standard curve. Absorbances were measured at 540 nm wavelength on Agilent 8453 UV/Visible Spectrophotometer (Agilent Technologies, USA).

### 4.5. Bacterial Viability after H_2_O_2_ and IAA Treatment

Selected bacterial cultures were incubated overnight at room temperature under shaking conditions (180 pm), with or without exogenous IAA (10 µg L^−1^) in liquid LB medium. These cultures were briefly treated with 10 µL of 30% hydrogen peroxide (H_2_O_2_). Cultures were then centrifuged at 10,000 rpm, and pellets were used to count colony-forming units (CFU). Pellets were diluted 10^1^, 10^2^, 10^3^ or 10^4^ times. Number of CFUs was used as the indicator of bacterial viability after oxidative stress. Untreated bacterial cultures with and without IAA were used as controls.

### 4.6. Biofilm Formation

Biofilm was quantified according to O’Toole (2010) [39]. Briefly, overnight cultures were diluted 1:100 in M9 medium with sucrose (20 mg L^−1^) or with IAA as the only energy source (10 µg L^−1^ and 10 mg L^−1^ final concentration). These bacterial suspensions were cultivated in a microtiter plate for 24 h at room temperature. The biofilm was determined by staining with crystal violet (0.1%) and measured spectrophotometrically at 550 nm wavelength.

### 4.7. Soft Agar Swim Assay

Chemotaxis of selected bacterial strains was tested in soft (0.3%) agar plates with 8.3 mM sodium-benzoate or 5 mM IAA as the sole carbon source, according to Scott et al. (2013) [38]. Briefly, soft M9 agar was prepared with M9 minimal salts and 0.3% agar. Soft agar plates with 8.3 mM sodium-benzoate as positive control and 5 mM IAA were prepared. Sodium benzoate served as a positive control as it is considered a common chemoattractant for *P. putida* and phylogenetically closely related strains [37]. A drop (5 µL) of fresh overnight cultures in LB medium was placed into the center of the soft agar. Plates were incubated for 24 h at +30 °C. Radial expansion of bacteria was measured (in millimeters). Plates without carbon source were used as negative controls.

### 4.8. Extract Preparation and HPLC-DAD Determination of IAA

Strains that were selected as previously described in Section 4.2 and Section 4.3 were cultivated for 3 days in 50 mL of LB medium at room temperature, under shaking conditions. After 3 days, the suspensions were centrifuged for 30 min at 10,000 rpm, and the supernatants were used for extraction in ethyl-acetate. Briefly, the supernatants were acidified with 1N HCl to pH 2.5 and mixed with ethyl-acetate 1:1. The aqueous phase was discarded, and the organic phase, containing IAA, was vacuum-evaporated at +70 °C until dry extract was formed. Prior to HPLC analysis, extracts were filtered through a 0.45 μm membrane filter (Captiva syringe filters, 0.45 mm, 13 mm, Agilent Technologies, Waldbronn, Germany).

Chromatographic analysis was carried out on Agilent series 1100 HPLC instrument (Agilent Technologies, USA), with a diode array detector, on a reverse-phase Zorbax SB-C18 (Agilent Technologies, USA) analytical column (250 mm × 4.6 mm i.d., 5 µm particle size) thermostated at 30 °C. The mobile phase consisted of solvent A (0.1% *v*/*v* solution of orthophosphoric acid in water) and solvent B (acetonitrile) using the gradient elution as follows: 80% A 0–10 min, 60% A 10–25 min, 0% A 25–30 min. The injection volume was 20 µL. Detection wavelengths were set at 218, 280 and 290 nm, and the flow rate was 1 mL min^−1^. Identification of IAA was confirmed by co-injection method using commercial standard of IAA (Sigma–Aldrich Chemie GmbH, Taufkirchen, Germany). Quantification was performed using standardized calibration curve of IAA (0.14–0.00875 mg mL^−1^). The content of IAA is presented as milligrams per gram of dry extract (mg g^−1^ DW) and as milligrams per optical density units (mg OD_600_^−1^) to account for differences in bacterial growth.

### 4.9. Co-Cultivation of Selected Bacterial Strains and Duckweeds

To test the effects of bacteria on plants, bacteria and plants were co-cultivated long-term in Murashige and Skoog medium, for 14 days. Previously sterilized and prepared duckweeds were transferred in sterile jars with Murashige and Skoog medium. Bacterial cultures were grown in LB medium (at +28 °C) under shaking conditions (180 rpm). Overnight cultures were centrifuged at 10,000 rpm and resuspended in sterile distilled water, then incubated with sterile duckweeds. Plants were kept at 24 ± 2 °C (under fluorescent light of 40 μmoL m^−2^ s^−1^ with 16 h light/8 h dark photoperiod). Bacteria/duckweed co-cultures were photographed every two days. Similarly, to test the effects of bacteria on plants cultivated in media supplemented with IAA, duckweeds were also cultivated with selected bacteria in the presence of 10 mg L^−1^ or 100 mg L^−1^ of exogenous IAA short-term, i.e., over 7 days. Images were analyzed in ImageJ (64-bit Java 8 version). Relative growth rate (RGR) was calculated according to the following formula:
(1)
(ln⁡x2−ln⁡x1)(T2−T1)

where *x* is the number of fronds at time points 1 and 2, and *T* is period of cultivation, in days. RGR is expressed in day^−1^ units.

Doubling time (DT) was calculated based on the following formula:
(2)
ln2RGR


DT was expressed in days.

### 4.10. Phylogenetic Analysis

DNA sequences (coding for 16S rRNA) of rhizosphere-associated bacterial strains were obtained as previously described [21]. Since the strains were previously identified based on 16S rRNA, these sequences were analyzed in order to obtain information regarding phylogenetic relationships between IAA-degrading and IAA-producing *Pseudomonas* strains. The evolutionary history was inferred using the neighbor-joining method [40]. The tree was drawn to scale, with branch lengths in the same units as those of the evolutionary distances used to infer the phylogenetic tree. The evolutionary distances were computed using the maximum composite likelihood method [41] and are in the units of the number of base substitutions per site. The number of bootstrap replications was 1000. This analysis involved 16S rRNA nucleotide sequences of the 4 selected bacterial strains and sequences of 8 phylogenetically closely related strains taken from the GenBank database (*P. putida* strains JCM 21368 and 1290, access. No. LC752231.1 and CP039371.1; *P. gessardii* strains 68soiLBA and P3 51 with access. No. LT38156.1 and LC717397.1; and 2 reference *P. oryzihabitans* strains, D84004.1 and LC191549.1; *P. yamanorum* strains 8H1 and LP2, access. No. NR178342 and KUT11080.1, respectively). All ambiguous positions were removed for each sequence pair (pairwise deletion option). Evolutionary analyses were conducted in MEGA software (version 11.0) [42].

### 4.11. Statistical Analysis and Graphical Presentation of the Results

Statistical analyses were performed using Statistica 10 software (StatSoft, Hamburg, Germany). Determination of all parameters was performed in three biological samples. In addition, the measurements for each sample were performed three times. All results are presented as means ± standard errors. Comparisons between means were made using Fisher’s LSD (least significant difference) post-hoc test calculated at a confidence level of *p* ≤ 0.05. The results were graphically presented using Microsoft Office Excel (2010).

## 5. Conclusions

To the best of our knowledge, this is the first study that reports strains of *P. oryzihabitans* D1-104/3, *P. putida* A3-104/5, *P. gessardii* C31-106/3 and *P. yamanorum* C44-104/1 from the rhizosphere of an aquatic plant, the common duckweed, as bacteria able to synthetize IAA and use it as a sole energy source. IAA has varying effects on the viability, motility and biofilm of these bacteria, which might be of importance in the natural environment where plants produce IAA that enhances the survival of bacteria, which might in turn confer adaptive benefits to plants. Dual IAA-producing/degrading ability is probably connected with the colonization of duckweeds and depends on exogenous IAA. Plant-growth-promoting effects depended on the presence of IAA, with *P. putida* A3-104/5, *P. gessardii* C31-106/3 and *P. yamanorum* C44-104/1 having a positive effect on the multiplication of duckweeds in the presence of IAA, reflected in significantly shorter doubling time compared to sterile duckweeds in an IAA-containing medium. Further research (e.g., involving insertional mutants for IAA production and degradation) would elucidate the exact mechanisms of action of these IAA-producing and -degrading strains on plants.

## Figures and Tables

**Figure 1 ijms-24-17207-f001:**
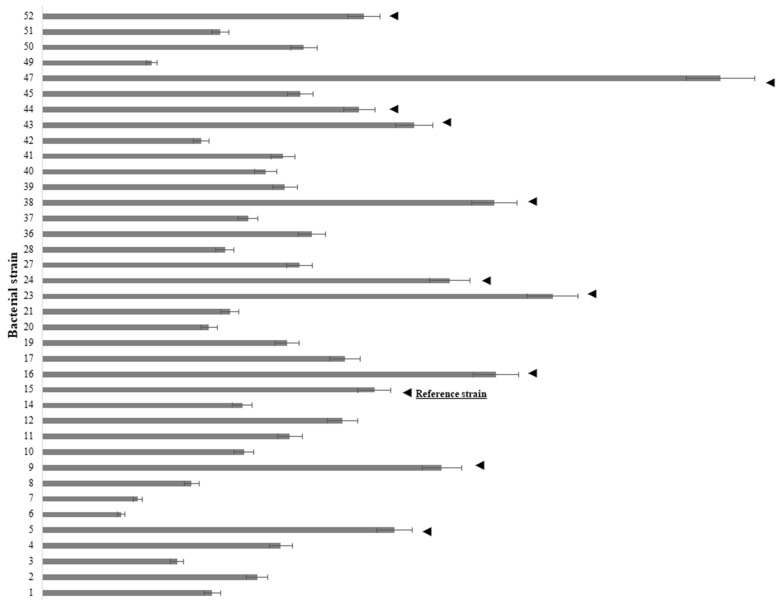
IAA production by bacterial strains isolated from the rhizosphere of duckweeds, first reported by Radulovic et al. (2019) [21]. *P. oryzihabitans* D1-104/3 (15, underlined) was used as a reference strain. All strains with production equal to or greater than (10.2 ± 0.3) mg L^−1^ (marked with ◄) were then transferred to minimal M9 medium with IAA as the sole carbon source. Bacterial strains are annotated with informal IDs for simplicity (15, 23, 38, 47, etc.) same as in our previous work [21].

**Figure 2 ijms-24-17207-f002:**
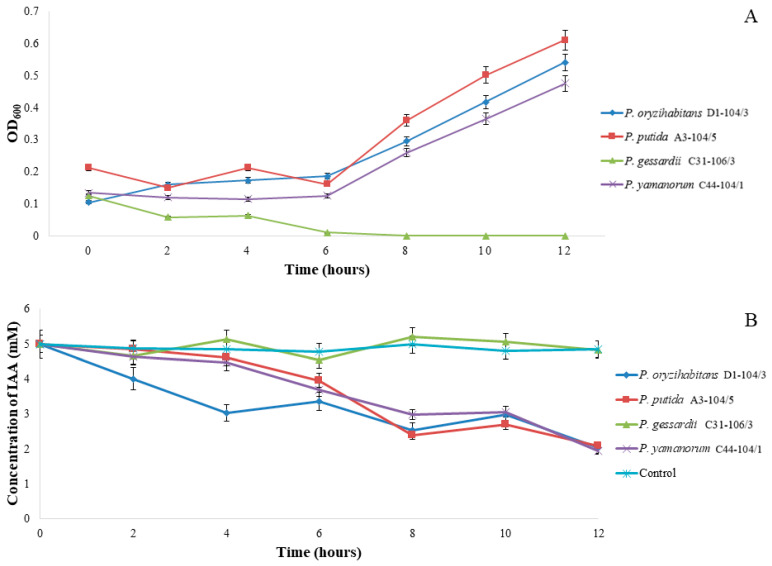
Growth of bacterial strains in (**A**) minimal M9 medium with IAA as the sole carbon source. (**B**) Degradation of IAA in minimal M9 medium with 5 mM of IAA as the sole carbon source, with sterile M9 with 5 mM IAA as the control.

**Figure 3 ijms-24-17207-f003:**
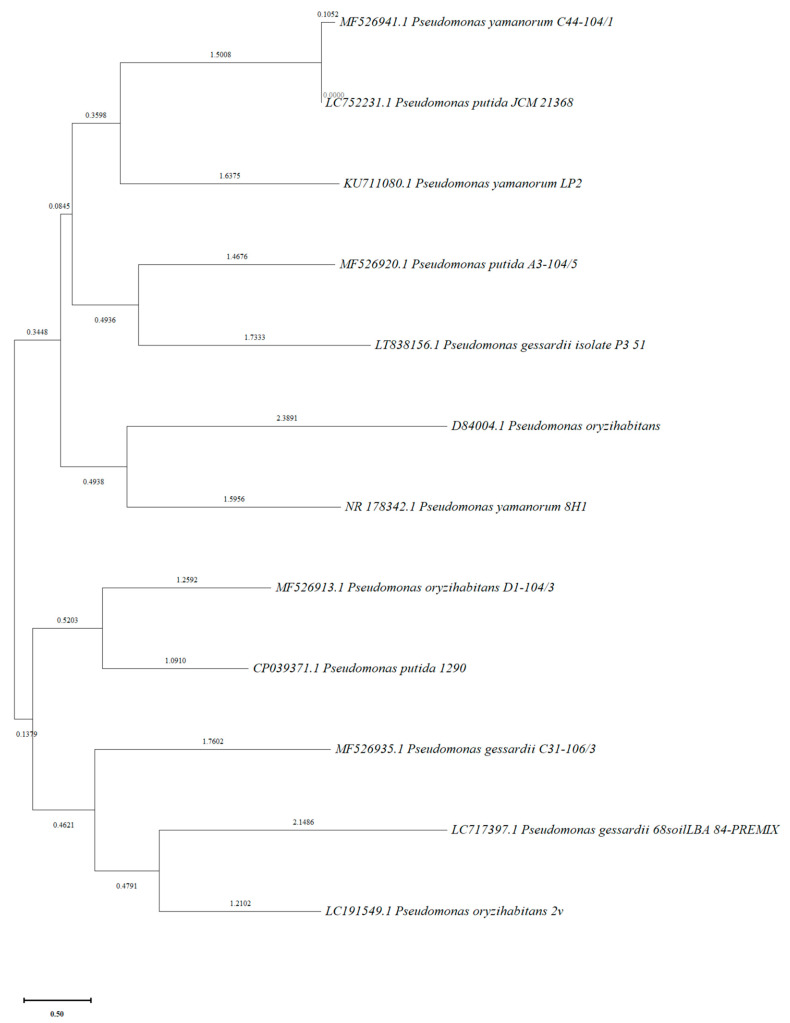
Phylogenetic tree generated by using the neighbor-joining method of analysis of 16S rRNA gene sequences of *Pseudomonas* strains including reference sequences from GenBank. Units of branch length are expressed in number of base substitutions per site. Bar represents the scale of genetic divergence.

**Figure 4 ijms-24-17207-f004:**
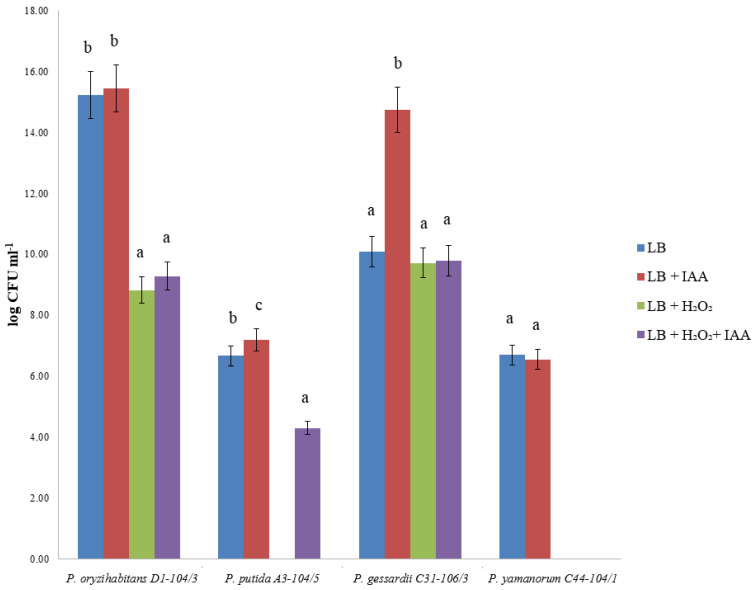
Bacterial viability after exposure to oxidative stress (H_2_O_2_ treatment). LB and LB + IAA represent bacterial colonies grown in LB nutrient medium with or without IAA but without subsequent exposure to oxidative stress. Data represent means ± standard errors. The effects of oxidative stress and IAA treatment on bacterial viability were evaluated using standard analysis of variance (ANOVA). Means marked with different letters are significantly different from control according to the LSD test (*p* ≤ 0.05).

**Figure 5 ijms-24-17207-f005:**
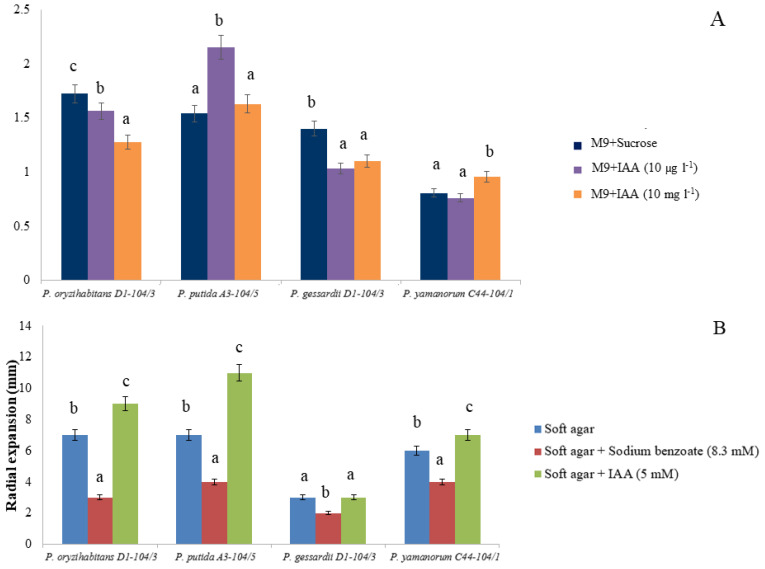
Biofilm production and chemotaxis. (**A**) Biofilm formation in M9 medium (with sucrose) and in IAA-supplemented M9 medium. (**B**) Results of soft agar swim assay. The effects of IAA on biofilm formation, as well as of sodium benzoate and IAA on bacterial radial expansion, were evaluated using standard analysis of variance (ANOVA). Means marked with different letters are significantly different from control according to the LSD test (*p* ≤ 0.05).

**Figure 6 ijms-24-17207-f006:**
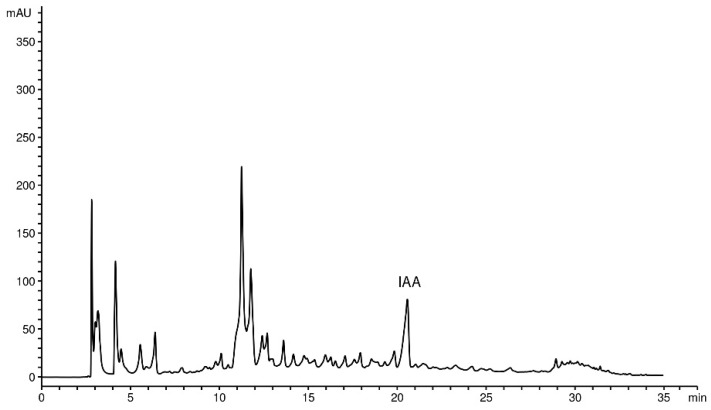
Representative chromatogram (λ = 280 nm) of ethyl-acetate extract derived from the supernatant of *P. oryzihabitans* D1-104/3 culture. Peak corresponding to IAA is annotated appropriately (‘IAA’). This peak appeared at retention time Rt = 20.56–20.59 min in all tested extracts.

**Figure 7 ijms-24-17207-f007:**
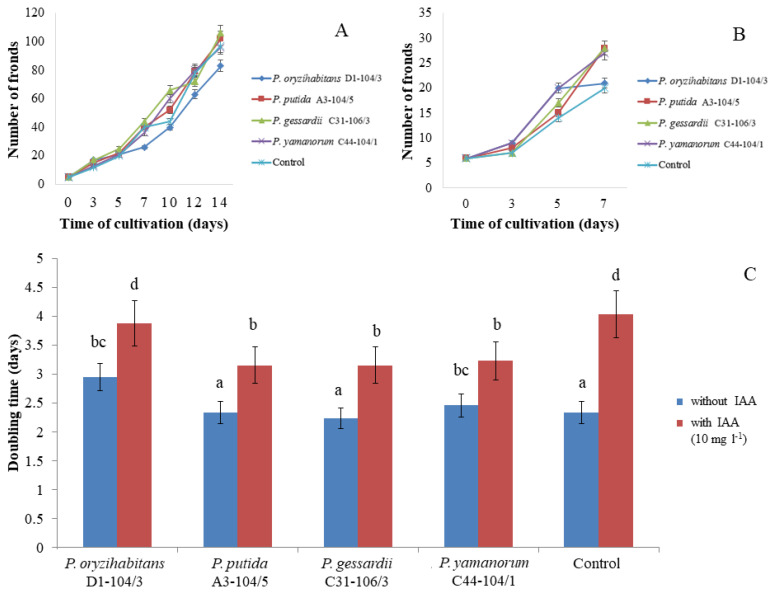
Co-cultivation of bacteria and duckweeds. (**A**) Growth of duckweeds (14 days) in MS medium without exogenous IAA. (**B**) Growth of duckweeds (7 days) in MS medium with exogenous IAA (10 mg L^−1^). (**C**) Doubling time (DT) in MS medium with and without exogenous IAA (10 mg L^−1^). The effects of IAA and bacteria on doubling time of duckweeds were evaluated using two-factor analysis of variance (ANOVA). Data represent means ± standard errors. Means marked with different letters are significantly different from control according to the LSD test and can be compared among each other (*p* ≤ 0.05).

**Table 1 ijms-24-17207-t001:** Information about selected IAA-producing and -utilizing strains. *P. oryzihabitans* D1-104/3, underlined, is used as a reference strain in this study.

Annotation	Strain	Species	Accession Number
15	D1-104/3	* Pseudomonas oryzihabitans *	MF526913
23	A3-104/5	*Pseudomonas putida*	MF526920
38	C31-106/3	*Pseudomonas gessardii*	MF526935
47	C44-104/1	*Pseudomonas yamanorum*	MF526941

**Table 2 ijms-24-17207-t002:** Growth parameters in minimal M9 medium with 5 mM of IAA as the sole carbon source.

Strain	Generation Time (h)	Growth Rate (h^−1^)	Relative Yield (OD_600_ Units Per 1 mmol of IAA)
*P. oryzihabitans* D1-104/3	7.64 ± 0.08	0.09 ± 0.01	0.08 ± 0.02
*P. putida* A3-104/5	5.60 ± 0.05	0.12 ± 0.02	0.05 ± 0.01
*P. gessardii* C31-106/3	N/A	N/A	N/A
*P. yamanorum* C44-104/1	6.58 ± 0.05	0.10 ± 0.01	0.11 ± 0.05

**Table 3 ijms-24-17207-t003:** IAA content produced by selected *Pseudomonas* strains, as determined in ethyl-acetate extract.

Strain	IAA Content (mg g^−1^ DW)	IAA Content (mg OD_600_^−1^)
*P. oryzihabitans* D1-104/3	7.41 ± 0.05	2.774 ± 0.005
*P. putida* A3-104/5	5.47 ± 0.07	1.936 ± 0.005
*P. gessardii* C31-106/3	1.12 ± 0.06	0.386 ± 0.009
*P. yamanorum* C44-104/1	1.16 ± 0.05	0.626 ± 0.005

## Data Availability

The data presented in this study are available on request from the corresponding author.

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
