# Peer review of "Characterization of the IAA-Producing and -Degrading Pseudomonas Strains Regulating Growth of the Common Duckweed (Lemna minor L.)"

_ijms, 2023, doi:10.3390/ijms242417207_

Round 1
Reviewer 1 Report
Comments and Suggestions for Authors
Overall, this article provides a thorough characterization of IAA producing and degrading Pseudomonas strains isolated from the rhizosphere of the common duckweed (Lemna minor). the article gives a more comprehensive picture of the rationale, experimental approaches, results, and discussion. However, there are still some areas that could be improved:
Introduction:
The introduction lacks clear objectives and hypotheses for the study. Specific aims should be stated at the end.
More background on the duckweed's biology and ecology would help justify why this aquatic plant was chosen.
The introduction could better highlight the novelty of studying these behaviors in aquatic plant-associated bacteria.
Materials and Methods:
The bacterial isolation and identification methods need to be described since this work builds on prior studies.
Details on duckweed growth conditions and experimental setup should be expanded.
Statistical tests used for data analysis are not specified and should be included.
Results:
Figures are referenced out of order between the two articles. The order and figure numbers should be checked.
Images of duckweed growth over time would provide visual evidence of effects described.
Statistical significance indicators should be added to graphs where appropriate.
The phylogenetic tree is referenced but not shown and should be included.
Discussion:
The discussion should provide more interpretation of the results and propose underlying mechanisms.
Limitations of the study approaches should be addressed along with future research directions.
Relating the findings back to the aquatic habitat would enhance the significance.
Overall, the combined manuscript provides a substantial body of work characterizing these novel IAA-interacting strains. However, strengthening the introduction, methods, results presentation, and discussion as outlined would improve the clarity and impact of the findings. Addressing these points would enhance this work as a cohesive, publication-quality manuscript.
Comments on the Quality of English LanguageOverall, this article provides a thorough characterization of IAA producing and degrading Pseudomonas strains isolated from the rhizosphere of the common duckweed (Lemna minor). the article gives a more comprehensive picture of the rationale, experimental approaches, results, and discussion. However, there are still some areas that could be improved:
Introduction:
The introduction lacks clear objectives and hypotheses for the study. Specific aims should be stated at the end.
More background on the duckweed's biology and ecology would help justify why this aquatic plant was chosen.
The introduction could better highlight the novelty of studying these behaviors in aquatic plant-associated bacteria.
Materials and Methods:
The bacterial isolation and identification methods need to be described since this work builds on prior studies.
Details on duckweed growth conditions and experimental setup should be expanded.
Statistical tests used for data analysis are not specified and should be included.
Results:
Figures are referenced out of order between the two articles. The order and figure numbers should be checked.
Images of duckweed growth over time would provide visual evidence of effects described.
Statistical significance indicators should be added to graphs where appropriate.
The phylogenetic tree is referenced but not shown and should be included.
Discussion:
The discussion should provide more interpretation of the results and propose underlying mechanisms.
Limitations of the study approaches should be addressed along with future research directions.
Relating the findings back to the aquatic habitat would enhance the significance.
Overall, the combined manuscript provides a substantial body of work characterizing these novel IAA-interacting strains. However, strengthening the introduction, methods, results presentation, and discussion as outlined would improve the clarity and impact of the findings. Addressing these points would enhance this work as a cohesive, publication-quality manuscript.
Author Response
Manuscript ijms-2661703
Response to Reviewer 1
Dear Sir/Madam,
We appreciate you allowing us to submit an updated version of our paper to International Journal of Molecular Sciences, which is titled “Characterization of the IAA Producing and Degrading Pseudomonas Strains Regulating Growth of the Common Duckweed (Lemna minor L.)”. We value the time and work you and editors have put into giving us insightful comments on our manuscript. We appreciate your thoughtful and thorough feedback on our work. Most of your ideas have been reflected in the improvements we have made. The changes made to the manuscript have been marked.
This is a point-by-point response to the issues and criticisms raised by the Reviewer 2:
Overall, this article provides a thorough characterization of IAA producing and degrading Pseudomonas strains isolated from the rhizosphere of the common duckweed (Lemna minor). the article gives a more comprehensive picture of the rationale, experimental approaches, results, and discussion. However, there are still some areas that could be improved:
Introduction:
The introduction lacks clear objectives and hypotheses for the study. Specific aims should be stated at the end.
Response: Thank you for your comment. We tried to better explain specific objectives in the last paragraph of introduction. (lines 96-107).
More background on the duckweed's biology and ecology would help justify why this aquatic plant was chosen.
Response: We added more information regarding duckweed’s biology (lines 73-76 and 89-94).
The introduction could better highlight the novelty of studying these behaviors in aquatic plant-associated bacteria.
Response: We provided a paragraph with a more detailed explanation (lines 48-56).
Materials and Methods:
The bacterial isolation and identification methods need to be described since this work builds on prior studies.
Response: We added reference were detailed isolation and identification protocol can be found (line 119).
Details on duckweed growth conditions and experimental setup should be expanded.
Response: We added more details on duckweed and bacterial co-cultivation and growth conditions (lines 204-209).
Statistical tests used for data analysis are not specified and should be included.
Response: We explained statistical tests in more detail (lines 241-243).
Results:
Figures are referenced out of order between the two articles. The order and figure numbers should be checked.
Response: We checked the issue, and each figure is placed after its call-out now.
Images of duckweed growth over time would provide visual evidence of effects described.
Response: We added representative images (Figure S3).
Statistical significance indicators should be added to graphs where appropriate.
Response: We added statistical significance indicators (Figures 4, 5 and 7).
The phylogenetic tree is referenced but not shown and should be included.
Response: The phylogenetic tree is shown in Figure 3.
Discussion:
The discussion should provide more interpretation of the results and propose underlying mechanisms.
Limitations of the study approaches should be addressed along with future research directions.
Relating the findings back to the aquatic habitat would enhance the significance.
Response: We revised the whole discussion section and provided a more detailed interpretation of results. We explained the study limitations at the end of the Discussion section (lines 535-542).
Overall, the combined manuscript provides a substantial body of work characterizing these novel IAA-interacting strains. However, strengthening the introduction, methods, results presentation, and discussion as outlined would improve the clarity and impact of the findings. Addressing these points would enhance this work as a cohesive, publication-quality manuscript.
Once again, thank you for your meaningful comments. We did our best to address the issues you raised and make improvements to our manuscript.

Reviewer 2 Report
Comments and Suggestions for Authors
General comments:
- Authors should consider making their figures more consistent with each other, especially for all the barplots that they present. This applies to bar colours, bar width, significance asterisks. The heterogeneity in all these plots show lack of care for the presentation of the results, and is at least distracting.
- Throughout the manuscript authors refer to 'strain' but often only include species name, and do not include the name of the strain that is being used. After all, authors are experimenting with specific strains, so they cannot conclude for the whole species.
Abstract
L17: since this is the first mention of the species name, do not abbreviate L. in L. minor.
General comment for Abstract: L18 until L30 seems to be a succession of highlights of the results, but there is not obvious connecting line for all these results. Authors should focus more on leading the reader through the goals of the work, and reduce the results highlights presented, so that the reader can understand why these experiments were made. This could be a challenge because IAA is related to many cellular responses, but ultimately it will improve the abstract.
Introduction
L72-73: no need to redefine IAA, it was already established at the start of the introduction.
L74: no need to say 'one' in 'one half'.
L86-88: authors are giving the idea of phylogenetic distance between the strains in the study, and relating it to IAA production. Authors then use the common phylogenetic marker 16S rRNA gene sequence to produce the phylogenetic analyses. But if these ideas are to be connected, shouldn't the authors also have sequenced and analysed the differences in the genes that code for IAA in all experimental strains?
L84-89: as mentioned above for the General comment for Abstract, it seems that the overall goal of the study remains confused. In this last paragraph of the Introduction, were often authors state the general goal that binds the study together (especially necessary when the experiments are very different in nature, which is the case here), the authors simply appear to state specific goals of some of the experiments. So, even at this point the manuscript is lacking a general goal that unites the work.
Material and Methods
L94: 'water' is tap water? Distilled water with nutrients? ...?
L100-101, L146: reference is not numbered.
L107-110: 'results were presented graphically' is not informative for methods. The rest of the paragraph is relevant only in the sense that authors kept the numbering done in a previous work, but that belongs in the interpretation of a Figure/Table, ie, in a legend, not in methods.
L114: is it relevant to say the accession number for the 16S of the strain? I suppose it would only be relevant if the accession number was generated in the scope of the current work, and even then, it belongs in a method section related to 16S rRNA gene sequencing.
L200-201: species name lacking italics.
L205: bootstrap analysis is performed, but not shown in the Figure in results.
L217: it is not clear why authors use 'paired t-test' in their analysis. According to methods so far, the samples appear to be independent (so, not paired) and there does not seem to be a case of 'before-and-after' type of dependency of the samples in the experiments performed. This should be made clear by the authors. Regarding the asterisks for significance, authors should state what is the level of significance related to one asterisk, to two asterisks, and so on; otherwise, it is useless to say here that they use asterisks. And in any way, if significance asterisks are used in Figures, then their legends should have the association of an asterisk and its significance level.
Results
L22-224, L234: reference is not numbered.
L234: if strain 15 is used as a reference, why does it have an asterisk mark? As I understand from the legend, the asterisks here do not refer to statistical significance, but to indicate a selection of strains that are used in downstream analyses. As such, authors should consider using another symbol (suggestion: arrows) because asterisks have a very specific meaning in scientific figures, especially in barplots.
Figure 1 - authors should consider remaking the figure with higher quality, and remove the asterisk (or eventually another symbol) box outlines.
Table 1 - 'Accession number' does not need the ':'. Also, as I understand, strain D1-104/3 is underlined because it was the reference strain in Figure 1. Accordingly, authors should include a phrase in the legend of this Table to elucidate this. Figures/Tables, together with their legends, should be interpretable as standalone objects in a scientific manuscript. The underline of the same strain also occurs in Table 2.
L266-269: It is not clear why authors chose those other 12 strains to be included in the phylogenetic tree, especially considering that Pseudomonas is a highly populated genus. Also, why did the authors choose to have an unrooted tree?
Figure 3 - authors must increase the quality of this Figure. It is not possible to make out the names. Additionally, the legend does not detail what the numbers on the phylogenetic tree refer to (although it is mentioned in methods, but Figures + legends should be standalone objects). Also, bootstrap was calculated in this analysis (according to methods) but the results are not presented in the Figure (if they are added, then it should also be evidenced in the legend). Additionally, authors say that the bar represents '50% sequence divergence' -- is this accurate?
L276-279: according to methods, strains were either treated as 'control' (without adding H2O2) or they were treated with H2O2 for oxidative stress. However, here the phrasing is confusing because it seems that strains were grown on IAA (equivalent to the 'control' mentioned in methods) and after were exposed to H2O2. Is this a before-and-after measurement? Or (and I believe this to be the case, looking at Figure 4), did the authors simply have independent experiments where biomass from strains were grown in IAA, and biomass was grown in exposure to H2O2? This should be clarified so that it is not questionable, and also because it will affect the kind of statistical test that is applied (paired vs not-paired for example). -- Edit: the discussion in lines L419-420 further confuses me in this aspect.
Figure 4 - authors refer in the legend 'significance was annotated with **' but there are no asterisks on the plot. Additionally, 'p=0.004' seems like a result of a single test, but it is not clear what that test was comparing (which groups? Which conditions? Which strain(s)?). Additionally, authors state that they used t-test to calculate significance, but it is not clear how this was made. T-tests compare two groups, and here there are four groups for each Pseudomonas strain. So how are the authors applying t-test to these data?
Figure 5 - authors refer that they performed paired t-test to these data, but again this does not make sense when there are three groups in analysis for each strain (in both 5A and 5B). Additionally, 'paired' t-test would imply dependent samples, or a before-and-after scenario, but according to methods the samples are independent. Is the 'paired t-test' is being used to make multiple comparisons after a significant ANOVA? This is not correct. Authors should review and clarify this. Also, 'control' in 5A and 'negative control' in 5B should be reviewed for consistency -- are they really different? Or both mean without addition of chemical?
Table 4 - 'Strain' does not need the ':'.
L331, L335, L339: Figures have until this point been referred as 'Figure', but here the authors abbreviated to 'Fig.'.
Figure 7 - plots C and D could be combined, using different colours for the different treatments. This way it would be easier to understand the changes between the conditions 'no IAA' and presence of IAA. Regarding the statistical analysis here, does it make sense to perform ANOVA for all strains in each condition separately (which is what I understand from the legend)? Or, would it make more sense to compare both conditions for each strain separately? After all, what scientific question are the authors trying to answer with this experiment?
Discussion
General comment 1 for Discussion: after reviewing the statistical methods applied in the results, authors will need to adjust their discussion and conclusions.
General comment 2 for Discussion: the discussion section might benefit from subsections (as were made in the methods and results sections) to separate the different topics that are relevant.
L383-385: where does the knowledge of P. gessardii being more genetically diverse come from? If it is from other studies, they should be cited. If authors are concluding this from the 16S rRNA gene-based phylogenetic tree, then they should state that it is only based on information of 16S rRNA gene sequence. Additionally, how are the authors able to link phylogenetic diversity of a 'strain' to very specific growth aspects? At most, authors could loosely speculate, because no experiments were done to link those two pieces of information, and correlation is not causation.
L389-390: the abbreviation for Luria-Bertani was already established.
L398-399: are the bacteria that produce and degrade IAA actually rare, or do authors just not make as many degradation tests as they do production tests? From my personal experience, it is routine to test if plant-associated species produce IAA, but rare to check if they degrade this phytohormone. -- Edit: the authors actually admit to this in lines L409-410.
L435, L461: reference is not numbered.
L455-458: endophytes colonize the internal tissues of plants without causing disease symptoms to the host. Some endophytes can enter plant cells, but most stay in the spaces between host cells. Here, did the authors mean 'plant tissue' instead of 'plant cell'? Additionally, if authors refer to this 'strain' as an endophyte, thus accepting that it came from a healthy host specimen upon its isolation (see endophyte definition above), then it does not make sense to speak of phytopathogenicity in the next phrase. If authors merely want to say that the 'strain' came from surface-sterilised tissues, this is not the same as saying 'endophyte' because it matter if those tissues came from a healthy host or not. Additionally, authors say that this 'species' is not classified as phytopathogenic, but phytopathogenicity of microbes can be strain-dependent, and even then, can depend on environmental (or experimental) conditions that will trigger the response of the bacteria towards the host. So this phrase needs to be reviewed at various levels for clarification purposes.
Conclusions
L479-481: authors should clarify what they mean by 'positive effect' in L481 because just above they refer to positive effect for 'growth rate' -- so what is the next 'positive effect' related to?
References
General note: there are too many inconsistencies in reference formatting (which again shows lack of care for the publication) to mention all. So, I compile a few examples:
- Last names and initials of authors are inconsistent (sometimes with dot, sometimes without; sometimes with comma, sometimes without).
- Year of publication sometimes is in bold face, others inside parenthesis, others outside parenthesis. Year is also sometimes before the publication title, sometimes after publication title.
- DOI links are sometimes as 'DOI: 10.111/....' others as 'https://doi.org/10.111/...'
- Sometimes Publication journals are abbreviated, sometimes not.
- Sometimes Publication issue numbers are in bold face, sometimes not.
- Sometimes Publication month is referred, sometimes not.
Supplementary Materials
- As was mentioned for the main manuscript, strain names should be specified.
- In the first phrase of 'Results' italics are missing for species name.
Author Response
Manuscript ijms-2661703
Response to Reviewer 2
Dear Sir/Madam,
We appreciate you allowing us to submit an updated version of our paper to International Journal of Molecular Sciences, which is titled: “Characterization of the IAA Producing and Degrading Pseudomonas Strains Regulating Growth of the Common Duckweed (Lemna minor L.)”. We value the time and work you and editors have put into giving us insightful comments on our manuscript. We appreciate your thoughtful and thorough feedback on our work. Most of your ideas have been reflected in the improvements we have made. The changes made to the manuscript have been marked.
This is a point-by-point response to the issues and criticisms raised by the Reviewer 2:
General comments:
- Authors should consider making their figures more consistent with each other, especially for all the barplots that they present. This applies to bar colours, bar width, significance asterisks. The heterogeneity in all these plots show lack of care for the presentation of the results, and is at least distracting.
Response: we thank you for your remark. The graphs and figures have been revised and updated.
- Throughout the manuscript authors refer to 'strain' but often only include species name, and do not include the name of the strain that is being used. After all, authors are experimenting with specific strains, so they cannot conclude for the whole species.
Response: We agree that it is incorrect to refer to strains, when only species is mentioned. We changed this throughout the manuscript.
Abstract
L17: since this is the first mention of the species name, do not abbreviate L. in L. minor.
General comment for Abstract: L18 until L30 seems to be a succession of highlights of the results, but there is not obvious connecting line for all these results. Authors should focus more on leading the reader through the goals of the work, and reduce the results highlights presented, so that the reader can understand why these experiments were made. This could be a challenge because IAA is related to many cellular responses, but ultimately it will improve the abstract.
Response: We thank you for your thoughtful remark. We made changes to the Abstract and hopefully clarified our aims. Please refer to Lines 14 – 28 in the new, revised version of our manuscript.
Introduction
L72-73: no need to redefine IAA, it was already established at the start of the introduction.
Response: Thank you, we revised it (line 79).
L74: no need to say 'one' in 'one half'.
Response: Corrected (line 80).
L86-88: authors are giving the idea of phylogenetic distance between the strains in the study, and relating it to IAA production. Authors then use the common phylogenetic marker 16S rRNA gene sequence to produce the phylogenetic analyses. But if these ideas are to be connected, shouldn't the authors also have sequenced and analysed the differences in the genes that code for IAA in all experimental strains?
Response: This is indeed a very valid point and in the future. However, for the scope and the goals of this paper, we chose 16S rRNA sequencing as a more robust, more reliable, and more used and tested gene marker for phylogenetic analysis.
L84-89: as mentioned above for the General comment for Abstract, it seems that the overall goal of the study remains confused. In this last paragraph of the Introduction, were often authors state the general goal that binds the study together (especially necessary when the experiments are very different in nature, which is the case here), the authors simply appear to state specific goals of some of the experiments. So, even at this point the manuscript is lacking a general goal that unites the work.
Response: Thank you for your remark. We agree that the Introduction lacks coherence. We improved the introduction to the best of our ability. Please refer to the new, revised version of the manuscript for all details, more specifically to lines 101 – 105.
Material and Methods
L94: 'water' is tap water? Distilled water with nutrients? ...?
Response: The water referred to is non-sterile tap water (line 111).
L100-101, L146: reference is not numbered.
Response: Reference is now added.
L107-110: 'results were presented graphically' is not informative for methods. The rest of the paragraph is relevant only in the sense that authors kept the numbering done in a previous work, but that belongs in the interpretation of a Figure/Table, ie, in a legend, not in methods.
Response: The paragraph is now changed, and the rest is transferred to the legend accompanying Figure 1.
L114: is it relevant to say the accession number for the 16S of the strain? I suppose it would only be relevant if the accession number was generated in the scope of the current work, and even then, it belongs in a method section related to 16S rRNA gene sequencing.
Response: We initially added this information for more clarity. But we agree, it can be regarded as superfluous. It is now deleted.
L200-201: species name lacking italics.
Response: Corrected.
L205: bootstrap analysis is performed, but not shown in the Figure in results.
Response: The ‘bootstrap value’ is incorrect and it should instead stand ‘the number of bootstrap replications’. This value was 1000.
L217: it is not clear why authors use 'paired t-test' in their analysis. According to methods so far, the samples appear to be independent (so, not paired) and there does not seem to be a case of 'before-and-after' type of dependency of the samples in the experiments performed. This should be made clear by the authors. Regarding the asterisks for significance, authors should state what is the level of significance related to one asterisk, to two asterisks, and so on; otherwise, it is useless to say here that they use asterisks. And in any way, if significance asterisks are used in Figures, then their legends should have the association of an asterisk and its significance level.
Response: We thank you for this comment. This is indeed a serious mistake on our part. We performed unpaired t-tests now and corrected our results accordingly.
Results
L22-224, L234: reference is not numbered.
Response: We revised it, thank you.
L234: if strain 15 is used as a reference, why does it have an asterisk mark? As I understand from the legend, the asterisks here do not refer to statistical significance, but to indicate a selection of strains that are used in downstream analyses. As such, authors should consider using another symbol (suggestion: arrows) because asterisks have a very specific meaning in scientific figures, especially in barplots.
Response: Indeed, it is not statistical significance in this case, but simply to show which strains were analyzed further. We agree with your remark, and we changed the design of the figure. Asterisk is substituted with an arrowhead.
Figure 1 - authors should consider remaking the figure with higher quality, and remove the asterisk (or eventually another symbol) box outlines.
Response: The figure is now changed.
Table 1 - 'Accession number' does not need the ':'. Also, as I understand, strain D1-104/3 is underlined because it was the reference strain in Figure 1. Accordingly, authors should include a phrase in the legend of this Table to elucidate this. Figures/Tables, together with their legends, should be interpretable as standalone objects in a scientific manuscript. The underline of the same strain also occurs in Table 2.
Response: Captions/legends were expanded with missing information according to your criticism. Please refer to lines 259 – 264.
L266-269: It is not clear why authors chose those other 12 strains to be included in the phylogenetic tree, especially considering that Pseudomonas is a highly populated genus. Also, why did the authors choose to have an unrooted tree?
Response: These taxa were chosen as the closest to the ones used in this study. We showed an unrooted phylogenetic tree since the relationships between the taxa are the main focus, not the evolutionary time and ancestry of the strains.
Figure 3 - authors must increase the quality of this Figure. It is not possible to make out the names. Additionally, the legend does not detail what the numbers on the phylogenetic tree refer to (although it is mentioned in methods, but Figures + legends should be standalone objects). Also, bootstrap was calculated in this analysis (according to methods) but the results are not presented in the Figure (if they are added, then it should also be evidenced in the legend). Additionally, authors say that the bar represents '50% sequence divergence' -- is this accurate?
Response: We agree with your statement re: the quality of this figure. Hopefully, we have improved it now. The caption is now expanded to include the explanation for the branches. As for the bootstrap value, as mentioned before, it should read ‘the no. of bootstrap replications’. '50% sequence divergence' is not accurate and should read ‘scale of genetic divergence’. Please refer to Lines 300 – 306.
L276-279: according to methods, strains were either treated as 'control' (without adding H2O2) or they were treated with H2O2 for oxidative stress. However, here the phrasing is confusing because it seems that strains were grown on IAA (equivalent to the 'control' mentioned in methods) and after were exposed to H2O2. Is this a before-and-after measurement? Or (and I believe this to be the case, looking at Figure 4), did the authors simply have independent experiments where biomass from strains were grown in IAA, and biomass was grown in exposure to H2O2? This should be clarified so that it is not questionable, and also because it will affect the kind of statistical test that is applied (paired vs not-paired for example). -- Edit: the discussion in lines L419-420 further confuses me in this aspect.
Response: Yes, these (control and control + IAA) were independent measurements. Control refers to bacterial cultures in nutrient medium without IAA, and without subsequent H2O2 exposure. Control + IAA refers to nutrient medium with IAA, without subsequent H2O2 exposure. Oxidative stress refers to cultures grown in nutrient medium without IAA, that were subsequently exposed to H2O2. Oxidative stress + IAA means that cultures were grown in IAA-supplemented medium, and then exposed to H2O2. We agree that the description was confusing, and we tried to correct this in the revised version. Please refer to Lines 307 – 313.
Figure 4 - authors refer in the legend 'significance was annotated with **' but there are no asterisks on the plot. Additionally, 'p=0.004' seems like a result of a single test, but it is not clear what that test was comparing (which groups? Which conditions? Which strain(s)?). Additionally, authors state that they used t-test to calculate significance, but it is not clear how this was made. T-tests compare two groups, and here there are four groups for each Pseudomonas strain. So how are the authors applying t-test to these data?
Response: We corrected this. We agree that ANOVA testing is not necessary here and is confusing, so we removed this from the text.
Figure 5 - authors refer that they performed paired t-test to these data, but again this does not make sense when there are three groups in analysis for each strain (in both 5A and 5B). Additionally, 'paired' t-test would imply dependent samples, or a before-and-after scenario, but according to methods the samples are independent. Is the 'paired t-test' is being used to make multiple comparisons after a significant ANOVA? This is not correct. Authors should review and clarify this. Also, 'control' in 5A and 'negative control' in 5B should be reviewed for consistency -- are they really different? Or both mean without addition of chemical?
Response: We thank you for your insightful comment. Again, we agree that ANOVA testing was incorrect. We re-tested with an unpaired t-test and revised the results accordingly. The control in Figure 5A represents control medium with sucrose as carbon source, whereas negative control in Figure 5B means control medium with no carbon source. Please refer to the revised Caption for Figure 5 (lines 350 – 358).
Table 4 - 'Strain' does not need the ':'.
Response: Corrected.
L331, L335, L339: Figures have until this point been referred as 'Figure', but here the authors abbreviated to 'Fig.'.
Response: We revised it in the whole manuscript.
Figure 7 - plots C and D could be combined, using different colours for the different treatments. This way it would be easier to understand the changes between the conditions 'no IAA' and presence of IAA. Regarding the statistical analysis here, does it make sense to perform ANOVA for all strains in each condition separately (which is what I understand from the legend)? Or, would it make more sense to compare both conditions for each strain separately? After all, what scientific question are the authors trying to answer with this experiment?
Response: We agree with your comments. ANOVA test misses the point here and is removed. Graphs are changed, as well as accompanying text and caption. Please refer to Lines 389 – 391. For the revised caption, please refer to Lines 393 – 401.
Discussion
General comment 1 for Discussion: after reviewing the statistical methods applied in the results, authors will need to adjust their discussion and conclusions.
General comment 2 for Discussion: the discussion section might benefit from subsections (as were made in the methods and results sections) to separate the different topics that are relevant.
L383-385: where does the knowledge of P. gessardii being more genetically diverse come from? If it is from other studies, they should be cited. If authors are concluding this from the 16S rRNA gene-based phylogenetic tree, then they should state that it is only based on information of 16S rRNA gene sequence.
Response: Corrected, please refer to lines 439 – 441.
Additionally, how are the authors able to link phylogenetic diversity of a 'strain' to very specific growth aspects? At most, authors could loosely speculate, because no experiments were done to link those two pieces of information, and correlation is not causation.
Response: We agree with this comment. It cannot be inferred from the data that such a causative connection exists. Please refer to revised paragraph (lines 439 – 446 in the revised version of the manuscript).
L389-390: the abbreviation for Luria-Bertani was already established.
Response: Corrected.
L398-399: are the bacteria that produce and degrade IAA actually rare, or do authors just not make as many degradation tests as they do production tests? From my personal experience, it is routine to test if plant-associated species produce IAA, but rare to check if they degrade this phytohormone. -- Edit: the authors actually admit to this in lines L409-410.
L435, L461: reference is not numbered.
Response: The references are now numbered in the whole manuscript.
L455-458: endophytes colonize the internal tissues of plants without causing disease symptoms to the host. Some endophytes can enter plant cells, but most stay in the spaces between host cells. Here, did the authors mean 'plant tissue' instead of 'plant cell'? Additionally, if authors refer to this 'strain' as an endophyte, thus accepting that it came from a healthy host specimen upon its isolation (see endophyte definition above), then it does not make sense to speak of phytopathogenicity in the next phrase. If authors merely want to say that the 'strain' came from surface-sterilised tissues, this is not the same as saying 'endophyte' because it matter if those tissues came from a healthy host or not. Additionally, authors say that this 'species' is not classified as phytopathogenic, but phytopathogenicity of microbes can be strain-dependent, and even then, can depend on environmental (or experimental) conditions that will trigger the response of the bacteria towards the host. So this phrase needs to be reviewed at various levels for clarification purposes.
Response: We agree that including hypothetical pathogenicity of this strain in this way is confusing. We changed this paragraph to clarify the negative effect this strain had on multiplication of duckweeds. Please refer to Lines 521 – 524.
Conclusions
L479-481: authors should clarify what they mean by 'positive effect' in L481 because just above they refer to positive effect for 'growth rate' -- so what is the next 'positive effect' related to?
Response: We agree that saying only ‘positive effect’ is unclear. We changed this sentence.
References
General note: there are too many inconsistencies in reference formatting (which again shows lack of care for the publication) to mention all. So, I compile a few examples:
- Last names and initials of authors are inconsistent (sometimes with dot, sometimes without; sometimes with comma, sometimes without).
- Year of publication sometimes is in bold face, others inside parenthesis, others outside parenthesis. Year is also sometimes before the publication title, sometimes after publication title.
- DOI links are sometimes as 'DOI: 10.111/....' others as 'https://doi.org/10.111/...'
- Sometimes Publication journals are abbreviated, sometimes not.
- Sometimes Publication issue numbers are in bold face, sometimes not.
- Sometimes Publication month is referred, sometimes not.
Response: We corrected all these issues with reference formatting.
Supplementary Materials
- As was mentioned for the main manuscript, strain names should be specified.
- In the first phrase of 'Results' italics are missing for species name.
Response: We corrected these issues with Supplementary Materials.
Once again, the Authors would like to thank Reviewer 2 for the effort and time dedicated to revision of this manuscript.

Round 2
Reviewer 2 Report
Comments and Suggestions for Authors
General comments:
- I have made this comment before, but I feel I need to make it again: "Authors should consider making their figures more consistent with each other, especially for all the barplots that they present. This applies to bar colours, bar width, significance asterisks. The heterogeneity in all these plots show lack of care for the presentation of the results, and is at least distracting." Authors still need to pay special attention to figure 5: A) had a yy axis line (also there are still lines from excel graph borders), while B) does not; 'control' in A) and 'negative control' in B) refer to different things, so the legend should be changed to reflect that; colours in A) and B) are the same but mean very different things, so authors are leading the reader to think that they are similar; bars in A) have spaces between them, while in B) they do not. --- All of these details make the paper less readable and may even induce errors while reading (the colour scheme situation).
- I have made this comment before, but I feel I need to make it again: "Throughout the manuscript authors refer to 'strain' but often only include species name, and do not include the name of the strain that is being used. After all, authors are experimenting with specific strains, so they cannot conclude for the whole species." Authors still need to change this in several instances, for example: figure 2, figure 4, figure 5, figure 7.
- Several aspects of the experimental and data analyses procedures remain unclear. Namely: number of replicates for each experiment; culture media used in several instances; what 'control' refers to in different assays/Figures; and finally, how was the statistical analysis done. This last point is very troubling to me. It seems like the authors just use t-test for everything and anything. Is it even applicable? Are the data normal and homoscedastic? Was that even tested? How are they applying t-test when they have >2 groups in the analysis?
Materials and methods:
- L229-232: The phrase 'statistical significance was annotated with *, **, etc' should be integrated with the new phrase where authors state what is the significance level linked to each annotation.
Results:
- Fig 1 caption: authors state that they have here the IAA production from bacterial strains -- and this would suffice. Authors also say that these strains come from designations from another paper (citation number 21), which is fine, but is information that belongs in methods (as in, where do these strains come from). The caption should be rephrased because, as is, one could ready it as 'the IAA production of these strains was previously determined in paper 21' which is not true.
- Table 1 - 'Accession number' does not need the ':' (was already a comment in the first review round; authors did not change).
- Figure 3 - The annotation '16S ribosomal RNA gene partial sequence' or variants of it, should be removed from the strain names to de-clutter the phylogenetic tree. The necessary annotations include species and strain name, and accession numbers.
- L296: '(...) as previously described' -- previously means in the methods? If so, there is no need to say that here. Or previously means in another paper? If so, authors need to cite it.
- Figure 4 (and corresponding text in the manuscript, including methods, where I'm only now realising that the authors do not say which culture medium was used) - If you have to explain at length what is meant by 'control' (non exposure to H2O2), maybe it would be better to just state the conditions used (this is especially true if 'control' means different things throughout the manuscript). For instance: blue would be 'nutrient medium'; red 'nutrient medium + IAA', green 'nutrient medium + H2O2', purple 'nutrient medium + IAA + H2O2'.
- Figure 4 (now related to stats) - I'm even more confused after the authors' response to this issue. Authors now say that they performed an 'unpaired t-test' with p<0.001 and they only have this level of significance above one bar for one strain. I remain confused regarding how the authors are performing the test(s). I have previously noted "it is not clear what that test was comparing (which groups? Which conditions? Which strain(s)?). Additionally, authors state that they used t-test to calculate significance, but it is not clear how this was made. T-tests compare two groups, and here there are four groups for each Pseudomonas strain. So how are the authors applying t-test to these data?" and the authors respond "We agree that ANOVA testing is not necessary here and is confusing, so we removed this from the text." but ANOVA was not mentioned anywhere before. Are (were) the authors performing one ANOVA (and what is the applied design?) for each strain? Is there enough replication (which is not stated in methods) of the assays? If ANOVA was performed, then why do the authors talk about t-tests?! This is all very confusing.
- Figure 5 - my comments from Figure 4 and also form the previous review round can be repeated here: authors should replace 'control' with actual conditions because using 'control' and 'negative control' just leads the reader in the wrong direction. The colours should not be the same in plots A and B, if they relate to different conditions, again misleading the reader. And there are the same issues as before for the statistical analyses: authors need to describe how the statistical analyses were made. How are the authors applying t-test to data where they have three groups?! Is it like in the legend (L335-336) 'IAA vs control'? (By the way authors here are stating the results in a figure caption, which is not correct) Then, what about the comparison with the sodium benzoate condition? Is it another t-test? Because that is not correct! (multiple comparisons without p-correction methods lead to an increased type 1 error.
- L334: should be 'assay', not 'essay'.
- L335: there are 2 spaces between 'difference' and 'between'.
- I have done this comment before, and the authors replied with 'corrected' but they did not correct. Table 4: 'Strain:' does not need the ':'.
- Figure 7 - authors should consider removing the boxes around plots A and B for a cleaner presentation of the data. In panel C, 'DT' in the colours legend should be removed because 'DT' is in the yy axis. From the figure caption, it seems that the authors went from using an ANOVA to compare DT without IAA between all microbiological conditions (4 strains and the no-inoculum control) to t-tests for the same 5-group comparison -- this is incorrect because t-test should only be used in 2 group comparisons (where the response variable follows the assumptions of the test). I have asked about this in the first round of reviews (and I ask again), because the goal of the analysis is not clear: either the authors want to know if duckweed doubling time differs in the absence of IAA for the different microbiological conditions (and analyse separately for the +IAA groups) OR the authors want to know, for each microbiological condition, does doubling time change when IAA is present in the medium? These very different scientific questions reflect different hypothesis, and different hypothesis tests. Authors should clarify what they are trying to understand from the data and perform the adequate analyses.
Discussion:
- General comment for discussion (a repetition): authors should revise the statistical analyses again, and then review the discussion accordingly.
- The authors responses refer to lines that are not correct in this portion of the review. Eg, lines 521-524 include supplementary materials descriptions, so not a part of the discussion.
Supplementary materials
- Authors still need to specify strain names in several instances: page 1 'Results' 1st paragraph; Fig S2 caption.
- There's a blank page at the end. Is it supposed to be this way, or should it be removed?
Author Response
Manuscript No. ijms-2661703
International Journal of Molecular Sciences (IJMS)
Dear Sir/Madam,
The Authors greatly appreciate your constructive and thorough criticism of our manuscript. We are grateful to Editors and Reviewers for the opportunity to resubmit our revised manuscript for consideration for publication in IJMS. The corrections reflect the remarks received from the Reviewer. This is a point-by-point response to the comments raised by the Reviewer.
General comments:
- I have made this comment before, but I feel I need to make it again: "Authors should consider making their figures more consistent with each other, especially for all the barplots that they present. This applies to bar colours, bar width, significance asterisks. The heterogeneity in all these plots show lack of care for the presentation of the results, and is at least distracting." Authors still need to pay special attention to figure 5: A) had a yy axis line (also there are still lines from excel graph borders), while B) does not; 'control' in A) and 'negative control' in B) refer to different things, so the legend should be changed to reflect that; colours in A) and B) are the same but mean very different things, so authors are leading the reader to think that they are similar; bars in A) have spaces between them, while in B) they do not. --- All of these details make the paper less readable and may even induce errors while reading (the colour scheme situation). We revised all figures as suggested.
- I have made this comment before, but I feel I need to make it again: "Throughout the manuscript authors refer to 'strain' but often only include species name, and do not include the name of the strain that is being used. After all, authors are experimenting with specific strains, so they cannot conclude for the whole species." Authors still need to change this in several instances, for example: figure 2, figure 4, figure 5, figure 7. We changed it.
- Several aspects of the experimental and data analyses procedures remain unclear. Namely: number of replicates for each experiment; culture media used in several instances; what 'control' refers to in different assays/Figures; and finally, how was the statistical analysis done. This last point is very troubling to me. It seems like the authors just use t-test for everything and anything. Is it even applicable? Are the data normal and homoscedastic? Was that even tested? How are they applying t-test when they have >2 groups in the analysis?
- We are very grateful to the Reviewer for these comments regarding data analyses and data presentation, so we tried to improve it. Accordingly, in the section 2.9. (Statistical analysis and graphical presentation of the results) we stated: “Statistical analyses were performed using Statistica 10 software (StatSoft, Hamburg, Germany). Determination of all parameters was done in three biological samples. In addition, the measurements for each sample were performed three times. All results are presented as means ± standard errors. Comparisons between means were made using Fisher’s LSD (least significant difference) post-hoc test calculated at a confidence level of p ≤ 0.05“. As you can see directly in the revised text manuscript we completely analyzed the obtained results and we hope that everything seems all right now.
Materials and methods:
-
- L229-232: The phrase 'statistical significance was annotated with *, **, etc' should be integrated with the new phrase where authors state what is the significance level linked to each annotation.
- We revised as it was suggested.
Results:
- Fig 1 caption: authors state that they have here the IAA production from bacterial strains -- and this would suffice. Authors also say that these strains come from designations from another paper (citation number 21), which is fine, but is information that belongs in methods (as in, where do these strains come from). The caption should be rephrased because, as is, one could ready it as 'the IAA production of these strains was previously determined in paper 21' which is not true. We revised the caption according to your comment.
- Table 1 - 'Accession number' does not need the ':' (was already a comment in the first review round; authors did not change). Corrected.
- Figure 3 - The annotation '16S ribosomal RNA gene partial sequence' or variants of it, should be removed from the strain names to de-clutter the phylogenetic tree. The necessary annotations include species and strain name, and accession numbers. We revised the figure according to your comment.
- L296: '(...) as previously described' -- previously means in the methods? If so, there is no need to say that here. Or previously means in another paper? If so, authors need to cite it. '(...) as previously described' means mentioned in the Material and Methods section. We remove this part from the sentence.
- Figure 4 (and corresponding text in the manuscript, including methods, where I'm only now realising that the authors do not say which culture medium was used) - If you have to explain at length what is meant by 'control' (non exposure to H2O2), maybe it would be better to just state the conditions used (this is especially true if 'control' means different things throughout the manuscript). For instance: blue would be 'nutrient medium'; red 'nutrient medium + IAA', green 'nutrient medium + H2O2', purple 'nutrient medium + IAA + H2O2'. We revised it per your suggestion.
- Figure 4 (now related to stats) - I'm even more confused after the authors' response to this issue. Authors now say that they performed an 'unpaired t-test' with p<0.001 and they only have this level of significance above one bar for one strain. I remain confused regarding how the authors are performing the test(s). I have previously noted "it is not clear what that test was comparing (which groups? Which conditions? Which strain(s)?). Additionally, authors state that they used t-test to calculate significance, but it is not clear how this was made. T-tests compare two groups, and here there are four groups for each Pseudomonas strain. So how are the authors applying t-test to these data?" and the authors respond "We agree that ANOVA testing is not necessary here and is confusing, so we removed this from the text." but ANOVA was not mentioned anywhere before. Are (were) the authors performing one ANOVA (and what is the applied design?) for each strain? Is there enough replication (which is not stated in methods) of the assays? If ANOVA was performed, then why do the authors talk about t-tests?! This is all very confusing.
- The authors are very grateful for this remark from the Reviewer. Accordingly, as previously mentioned, we completely re-analysed the results obtained in this revised version of work. At first, we thought that t-test should be completely enough but based on your remarks considering statistics we realized that our explanation what we analysed and what we compared was not clear enough and even not maybe appropriate enough. So in the figure legends (Figures 4 and 5) we added completely new sentences considering applied statistical analyses. For the Figure 7 we applied two-factorial ANOVA (stated in the figure legend). Also, on all presented bar graphs different letters are added and they are indicating the significant different means in comparison to appropriate control values.
- Figure 5 - my comments from Figure 4 and also form the previous review round can be repeated here: authors should replace 'control' with actual conditions because using 'control' and 'negative control' just leads the reader in the wrong direction. The colours should not be the same in plots A and B, if they relate to different conditions, again misleading the reader. And there are the same issues as before for the statistical analyses: authors need to describe how the statistical analyses were made. How are the authors applying t-test to data where they have three groups?! Is it like in the legend (L335-336) 'IAA vs control'? (By the way authors here are stating the results in a figure caption, which is not correct) Then, what about the comparison with the sodium benzoate condition? Is it another t-test? Because that is not correct! (multiple comparisons without p-correction methods lead to an increased type 1 error.
- Statistical issues regarding Figure 5 are solved the same way as for the Figure 4. The standard one-way ANOVA is applied and the results of statistical analysis are presented in the Figure. The explanation of presented statistics is also stated in figure legend.
- L334: should be 'assay', not 'essay'. Corrected.
- L335: there are 2 spaces between 'difference' and 'between'. Corrected.
- I have done this comment before, and the authors replied with 'corrected' but they did not correct. Table 4: 'Strain:' does not need the ':'. Corrected.
- Figure 7 - authors should consider removing the boxes around plots A and B for a cleaner presentation of the data. In panel C, 'DT' in the colours legend should be removed because 'DT' is in the yy axis. Corrected
- From the figure caption, it seems that the authors went from using an ANOVA to compare DT without IAA between all microbiological conditions (4 strains and the no-inoculum control) to t-tests for the same 5-group comparison -- this is incorrect because t-test should only be used in 2 group comparisons (where the response variable follows the assumptions of the test). I have asked about this in the first round of reviews (and I ask again), because the goal of the analysis is not clear: either the authors want to know if duckweed doubling time differs in the absence of IAA for the different microbiological conditions (and analyse separately for the +IAA groups) OR the authors want to know, for each microbiological condition, does doubling time change when IAA is present in the medium? These very different scientific questions reflect different hypothesis, and different hypothesis tests. Authors should clarify what they are trying to understand from the data and perform the adequate analyses.
- Considering statistics in Figure 7, please refer to the answer related to the Figure 4.
Discussion:
- General comment for discussion (a repetition): authors should revise the statistical analyses again, and then review the discussion accordingly.
- Suggestion accepted. We revise statistical analysis and corrected discussion accordingly.
- The authors responses refer to lines that are not correct in this portion of the review. Eg, lines 521-524 include supplementary materials descriptions, so not a part of the discussion.
Supplementary materials
- Authors still need to specify strain names in several instances: page 1 'Results' 1st paragraph; Fig S2 caption.
- Revised.
- There's a blank page at the end. Is it supposed to be this way, or should it be removed? Removed.
Again, the authors would like to express their gratitude to the Reviewer for their thorough and detailed remarks on our manuscript.
